# Reactive Extrusion as a Pretreatment in Cassava (*Manihot esculenta* Crantz) and Pea (*Pisum sativum* L.) Starches to Improve Spinnability Properties for Obtaining Fibers

**DOI:** 10.3390/molecules27185944

**Published:** 2022-09-13

**Authors:** David Tochihuitl-Vázquez, Rafael Ramírez-Bon, José Martín Yáñez-Limón, Fernando Martínez-Bustos

**Affiliations:** Centro de Investigación y de Estudios Avanzados del Instituto Politécnico Nacional (CINVESTAV-Unidad Querétaro), Libramiento Norponiente 2000, Fraccionamiento Real de Juriquilla, Querétaro 76230, Mexico

**Keywords:** electrospinning, fibers, pea starch, cassava starch, reactive extrusion, entanglement concentration

## Abstract

Starch is a biocompatible and economical biopolymer in which interest has been shown in obtaining electrospun fibers. This research reports that cassava (CEX) and pea (PEX) starches pretreated by means of reactive extrusion (REX) improved the starches rheological properties and the availability of amylose to obtain fibers. Solutions of CEX and PEX (30–36% *w*/*v*) in 38% *v/v* formic acid were prepared and the rheological properties and electrospinability were studied. The rheological values indicated that to obtain continuous fibers without beads, the entanglement concentration (Ce) must be 1.20 and 1.25 times the concentration of CEX and PEX, respectively. In CEX, a higher amylose content and lower viscosity were obtained than in PEX, which resulted in a greater range of concentrations (32–36% *w*/*v*) to obtain continuous fibers without beads with average diameters ranging from 316 ± 65 nm to 394 ± 102 nm. In PEX, continuous fibers without beads were obtained only at 34% *w/v* with an average diameter of 170 ± 49 nm. This study showed that starches (20–35% amylose) pretreated through REX exhibited electrospinning properties to obtain fibers, opening the opportunity to expand their use in food, environmental, biosensor, and biomedical applications, as vehicles for the administration of bioactive compounds.

## 1. Introduction

Electrospinning is a versatile and cost-effective electrodynamic technique, capable of producing micro- to nanoscale fibers with structural and functional properties such as high porosity, high surface area/volume ratio, and encapsulation efficiency that can be obtained from biopolymers, synthetic polymers, or a combination of both [1]. Due to their renewability, biodegradability, biocompatibility, and low cost, electrospun fibers from biopolymers have aroused interest in applications that include the food [2], environment [3], biosensors [4], and biomedical [5] sectors as supply-systems bioactive compounds.

Starch is one of the most abundant biopolymers in nature. It consists of glucose molecules linked by glycosidic bonds, whose classification is based on their content of amylose and amylopectin, in waxy starches (>90% amylopectin), high in amylose (>40% amylose), and in normal starches (20–35% amylose) [6]. It can be obtained from various vegetable sources such as corn, wheat, potato, rice, pea, and cassava [7,8].

Compared with potato, wheat, and corn starch, cassava and pea starches are in high demand due to their low cost and unique functional characteristics. The production of cassava (*Manihot esculenta* Crantz) for starch is increasing rapidly (by over 3% per year) and accounts for about 7% of the starch produced globally. Cassava roots are rich in starch (approximately 84.5%) and have an amylose content that ranges between 15.2% and 26.5%. The increasing demand is due to its special characteristics, such as its being allergy-free, its having a blunt taste, clarity, and freeze–thaw stability, which are very attractive to the food and non-food industries [9].

Pea (*Pisum sativum* L.) is one of the most important food crops in agricultural grain production, and around 25 million hectares of peas are planted each year worldwide. Pea starch, a by-product of pea protein extraction, accounts for about 40% of the dry weight of the seeds and is considered an inexpensive source of starch with an amylose content that ranges between 29.4% and 65.0%. In addition, it is often used in food formulations due to its slow digestion and high content of resistant starch [10,11].

Several researchers have reported that obtaining fibers from starch via electrospinning depends on the amylose content and the rheological properties of the solution, which facilitate the interactions among the chains to achieve the required level of molecular entanglement.

Consequently, the production of electrospun fibers from starch has focused mainly on high amylose starches and the use of high concentrations of strong polar solvents or mixed solvents (dimethyl sulfoxide or formic acid) to dissolve the starch. For example, Fonseca et al. [12] obtained electrospun fibers from corn starch with an amylose content of 70% in formic acid (75% *v*/*v*) during long aging times (24, 48, and 72 h) to dissolve the starch. Lancuški et al. [13], reported that solutions of Hylon starch (70% amylose) in 100% and 90% *v/v* formic acid formed uniform fibers. Similarly, Kong and Ziegler [14] electrospun high amylose corn starch (80% *w*/*w*) in dimethyl sulfoxide (95% and 100% *v*/*v*).

Unfortunately, the high cost of high amylose starches and the use of high concentrations of strong polar solvents limit the production of fibers by electrospinning. To obtain electrospun starch fibers with economic viability on a commercial scale, approaches are needed that allow normal starch to be processed and to overcome the limitations that it entertains in its native form. For example, we find it has low solubility in water and high viscosity in aqueous solvents, due to the strong hydrogen bonds between the starch chains, a high degree of polymerization, and complex semi-crystalline structures that generate an effective volume fraction much higher than its actual volume fraction [15].

One strategy to overcome these drawbacks and improve the physicochemical properties of starch is through physical, chemical, and enzymatic treatments. Physical treatments are more widely accepted as an ecological, profitable, and highly efficient alternative [16], and among the physical treatments REX is being increasingly adopted due to its being a process with a low operating cost. In addition, its adoption may encompass the possibility of a continuous process that implies high shear and rapid heat transfer and efficient mixing, with significant changes in the properties of starch, such as partial gelatinization, fusion, fragmentation, loss of crystallinity, and a decrease in molecular order [17].

Researchers [18,19] have reported that starch under REX conditions exhibits a decrease in molecular order, allowing the improvement of the starch’s end-use properties by breaking the hydrogen bonds between straight-chain amylose and branched-chain amylopectin, with amylopectin more susceptible to changes in size distribution. For example, Lai and Kokini [20] found that starch granules after extrusion were fragmented by high shear and that they exhibited a decrease in viscosity and an increase in solubility in solution.

Fasheun et al. [21] applied REX to a mixture of cassava starch with sugarcane bagasse and reported that the physicochemical properties, such as the solubility index and swelling power, improved.

Therefore, in this study, our objective was to apply REX in cassava (NCS) and pea (NPS) starches to improve the physicochemical and rheological properties for obtaining fibers by the electrospinning technique. Electrospun fibers were successfully manufactured from CEX and PEX starches and electrospun fibers were characterized.

Additionally, the spinnability and rheological properties of starch solutions were studied. Among the rheological properties, the semi-dilute unentangled rate, the semi-dilute entangled rate, and the Ce were identified for obtaining electrospun fibers. These results provide the opportunity to expand the use of normal starches in obtaining electrospun fibers as a vehicle for bioactive molecules. To our knowledge, there are no reports in the literature that use REX as a pretreatment in starches for the manufacture of fibers by means of the electrospinning technique.

## 2. Results and Discussion

### 2.1. Morphological Characterization and Particle-Size Distribution of Starches

To observe the effect of REX on starches, the microstructure of CEX and PEX was analyzed by scanning electron microscopy (SEM) and was compared with NCS and NPS. Figure 1a shows that the NCS granule is intact and has round ellipsoidal-like morphology, with a truncated outer surface with a mean size of 12.7 µm. Similar results were reported for tapioca starch [22]. Figure 1b reveals that the NPS granule presented elliptical and spherical morphology with a smooth external surface and no cracks, with an average size of 20.3 µm. These results are in agreement with the information reported by Zhang et al. [23].

The CEX and PEX starches (Figure 1c,d) presented damaged granular structures, with irregular fractures associated with the fusion of the crystallites due to the high temperature and the shear force applied during the REX process; these could favor rapid diffusion of the solvent into starches during the preparation of solutions.

Figure 2 reveals that CEX presented a higher degradation rate than PEX. This is attributed to the reduced particle-size distribution (PSD) at the maximal peak and bimodal-size distribution with values of 183.2 nm and 3438 nm, due to the granular fractionation of the starch attributed to its size and to the damage of amylopectin molecules due in turn to its high molecular weight; meanwhile, PEX exhibited broad distribution in the maximal peak and in three size-distribution peaks (4036 nm, 1124 nm, and 123.1 nm), indicating minor degradation and a change in size from larger to smaller. However, PEX demonstrated a PSD with a maximal peak of 123.1 nm, smaller than that of CEX (183.1 nm). This can be attributed to the cleavage of the low-molecular-weight amylopectin backbone of PEX comprising a small number of long branches because of starch biosynthesis in the amylose/amylopectin ratio [24].

### 2.2. Physicochemical Properties of Starches

The values of the apparent amylose content (AAC), solubility (S), and swelling power (SP) of the NCS, NPS, CEX, and PEX starches are presented in Table 1. The AAC values obtained in the NCS and NPS starches were 21.41 ± 0.06% and 32.24 ± 0.12%, respectively, indicating that they belong to the classification of normal starches. Similar values of amylose content have been reported in cassava [25] and pea starches [10]. The CEX and PEX starches demonstrated significantly higher values (*p* < 0.05) of ACC, S, and SP than the NPS and NCS starches. This is due to the partial gelatinization of the starch that led to the release of amylose and changes in the structure of the starch’s amylopectin during the REX process [26,27]. A similar effect was found in rice and waxy rice flours after the extrusion process [28].

In this study, the AAC values obtained in CEX (49.02 ± 0.13) and PEX (43.36 ± 0.13) permitted sufficient molecular entanglement in electrospinning solutions to obtain fibers with continuous morphology and without beads. Other investigators confirmed that high amylose content in starches is essential for fiber formation due to their ease of molecular entanglement in the electrospinning solution [29]. For example, Fonseca et al. [12] obtained fibers with different morphologies from Hylon starches with amylose contents of 70% and 55%, using aqueous formic acid (75% *v*/*v*) as a solvent to induce chemical gelatinization during an aging time of 24, 48, and 72 h.

The starches CEX and PEX revealed that when they were pretreated by REX, they improved their S and SP due to the breaking of hydrogen bonds and covalent bonds between molecules [16], which facilitated the diffusion of the aqueous solvent toward the starches in electrospinning solutions and the use of a higher proportion of water in the solvent system (water/formic acid).

The values of AAC, S, and SP were significantly higher (*p* < 0.05) in CEX than in PEX.

This can be explained as due to the molecular size of the amylopectin present in cassava, which is more susceptible to REX conditions [18,19], therefore allowing for the improvement of the rheological properties in the electrospinning solution and the amylose content necessary for the chains to interact with each other.

On the other hand, the values of S and SP were lower in NPS than in NCS. This is attributed to the characteristic of low solubility and to the higher content of amylose present in pea starches, which restricts the swelling of starch granules during gelatinization [30].

### 2.3. Rheological Properties and Electrospinability of Starches

Rheological properties such as viscosity and molecular entanglements in polymer solutions exert a great influence on electrospinning [14]. For example, low viscosity leads to the formation of beads or droplets during electrospinning; otherwise, the flow will cause a blockage in the capillary [27], while sufficient molecular entanglement is necessary to stabilize the long-range network and facilitate fiber formation from the ejected polymer solutions [31]. The flow curves of the CEX and PEX solutions at concentrations ranging from 0.1–38% (*w*/*v*) are depicted in Figure 3a,b. It was observed that the viscosity of the solutions increased as the concentration of CEX and PEX increased; this was more notable in PEX due to its granulometric heterogeneity, generated by the lesser amount of damage undergone during REX that may contribute to a greater hydrodynamic volume and therefore high resistance to flow. When the concentrations of CEX and PEX were low they exhibited a trend toward Newtonian behavior, where the concentration of the starches increased by more than 24% and shear thinning was shown, resulting in pseudoplastic behavior that was more evident in PEX.

Figure 4a,b show the adjusted slopes of the semi-dilute unentangled rate, the semi-dilute entangled rate, and the Ce for the CEX and PEX solutions. Theoretical predictions suggest that in the dilute regime *n* is 1.0, while in the semi-dilute unentangled regime *n* is 1.25 when the C of the polymer is greater than C* and less than Ce. For the semi-dilute entangled regime *n* is 4.8, that is, the C* is less than the C of the polymer for linear and neutral polymer solutions.

The semi-dilute unentangled rate, the semi-dilute entangled rate, and the Ce for electrospun fibers were identified [31]. It was observed that the value of the diluted regime of CEX and PEX was 0.63 and 0.66, respectively, indicating that there was no overlap of polymeric chains according to the theoretical prediction [32].

The semi-dilute entangled value of the CEX and PEX starches was 3.16 and 3.24, respectively, which was higher than the theoretically predicted *n* = 1.25 for linear and neutral polymers; this demonstrated a stronger concentration dependence than the theoretical prediction. The latter could be attributed to a reduced viscosity due to the decrease in the molecular weight of amylopectin by means of the REX process and redistribution in portions of the fractions with different degrees of polymerization.

In the semi-dilute entangled regime, concentration dependencies were 4.45 and 4.57 for CEX and PEX, respectively. These values are near the theoretical prediction reported (*n* = 4.8) for linear polymers, indicating the presence of molecular entanglements in the concentration of starches with a strong interaction, possibly attributed to a greater presence of linear amylose molecules that allow the overlap and entanglement of the chains [32,33].

Kong and Ziegler [14] reported similar values for Gelose 80 in 75% and 70% dimethyl sulfoxide aqueous solutions. However, in these solutions the starch could not be completely dissolved due to the significant increase in viscosity.

Likewise, a lower concentration-dependence value was observed in CEX that, despite its having a greater number of amylose chains that strongly contribute to the extended coils, it exerted an influence on the existence of chains with lower molecular weight that translates into a lower hydrodynamic volume in the polymer solution. Therefore, Ce was defined as the point at which the molecular chains significantly overlap to form interlocking couplings. Ce values were 28.14% and 27.44% for the CEX and PEX solutions, respectively. It was observed that the Ce value decreased in PEX, and this could be due to a higher molecular weight of the chains responsible for molecular entanglement; therefore, these require a lower concentration to become entangled. Furthermore, these results were much higher than the values reported for solutions of amylose and amylopectin in formic acid [34], suggesting that CEX and PEX starches must be more concentrated to establish significant molecular entanglement in solutions with higher ratios of water. If we consider the critical entanglement concentration (C**) at which fibers without defects begin to form, C**/Ce values can be obtained for starches in 38% (*v*/*v*) formic acid. These values are 1.14 and 1.25 for CEX and PEX, respectively. Our findings are consistent with reported C**/Ce values for without beads electrospun fibers for Gelosa 80 [14].

In this study, the solutions of CEX and PEX at 38% could not be electrospun, because in PEX, the electrospinning jet did not form, and in CEX the jet was unstable. Therefore, electrospinning solutions were considered within a concentration range of 30–36%.

These solutions exhibited non-Newtonian behavior where, as the shear rate increased, the apparent viscosity (*η*_app_) decreased (Figure 3a,b) with values of *n* < 1 from 0.95 ± 0.00 Pa·s to 0.85 ± 0.00 Pa·s (Table 2) described by the Herschel-Bulkley model. Similar values of *n* < 1 were reported in solutions of pea flour, hydroxypropyl methyl-cellulose, and polyethylene oxide (when the pH was increased in electrospinning solutions) to obtain nanofibers by electrospinning [35]. Other researchers [12] studied the behavior of soluble potato-starch solutions in formic acid employing the Herschel–Bulkley model and reported similar values of *n* < 1.

On the other hand, it is shown that by increasing the concentration from 30 to 36% in the electrospinning solutions, the ηapp increased from 0.72–1.90 Pa·s for CEX and from 1.15–2.62 Pa·s for PEX (Figure 3a,b). The 38% CEX solutions demonstrated better stability of the electrospun jet and the production of continuous fibers without beads (Figure 5a). In addition, the suitable viscosity for electrospun continuous fibers without beads for each polymer was of ~1.11–1.90 Pa·s for CEX and ~2.01 Pa·s for PEX.

Obtaining fibers without beads in PEX was limited to a specific viscosity, because a lower amylose content is sufficient for molecular entanglement at a given concentration. A similar trend was reported for a solution of Hylon VII starch in 95% dimethyl sulfoxide when it was electrospun at concentrations ranging from 8–20% (*w*/*v*), and this range became smaller as the amylose content of the starch decreased [14]. It was also observed that the *η*_app_ in PEX and CEX solutions is close to that reported in obtaining starch fibers from solutions of Hylon VII (15% *w*/*v*) aged from 24–48 h in formic acid with a viscosity range from 1.7–0.9 Pa·s [12].

Unlike what was reported in Hylon VII starch solutions at 17% *w/v* in formic acid at 100% and 90% *v/v* aged with viscosity values of 0.79 and 0.74 Pa·s, respectively, which showed homogeneous fibers without beads [13].

In this study, we observed that the viscosity of the CEX and PEX solutions (Figure 3a,b) exerted a greater influence on the diameter of the fibers (Table 3) than the electrical conductivity. The values of the consistency coefficient (K) of CEX and PEX (Table 2) entertained a significant difference (*p* < 0.05). As the concentration of PEX in the solutions increased, a higher viscosity was observed in a range of 81.14 ± 0.01 to 2.37 ± 0.03 Pa·s*^n^*; which could be attributed to the particle size; that is, to a certain concentration of starch where the smaller the particle size, the more particles there are in the solution, and consequently more interaction between the particles. In a previous study [35], pea-flour solutions for obtaining nanofibers showed values of K similar to those obtained in PEX and CEX solutions, while the elastic limit in these solutions did not reveal any significant difference (*p* > 0.05), implying low resistance to the initial electrospinning flow.

### 2.4. Electrical Conductivity of Starch Solution

Electrical conductivity determines the ability of a material to allow the passage of electrical current and to move towards the surface of the hanging droplet, in order to generate an electrostatic repulsive force that is critical to initiate the jet during electrospinning [36]. Therefore, electrical conductivity is essential in the formation of electrospun fibers. Increasing electrical conductivity in electrospinning solutions could improve fiber uniformity and decrease bead generation [37]. Table 3 shows that the solutions of CEX and PEX in concentrations of 30% and 36% presented a significant difference (*p* < 0.05), with an increase in electrical conductivity in the 30% concentration with values of 3.41 ± 0.09 mS/cm and 3.46 ± 0.33 mS/cm for CEX and PEX, respectively. However, at this concentration, the electrical conductivity did not influence the uniformity of the fibers. Thus, it was correlated with poor consistency in the morphology of the fibers (Figure 4g,h), while in the concentration at 36% of CEX and PEX low electrical-conductivity values of 2.77 ± 0.20 mS/cm and 2.58 ± 0.24 mS/cm, respectively, were obtained, with uniform morphology in the CEX and PEX fibers; this is possibly attributable to the increase in free hydroxyl groups.

In addition, in this study it was observed that the range of electrical conductivity was correlated with the uniformity of the fibers without beads: for CEX this was 2.77 ± 0.20 to 2.97 ± 0.27 mS/cm, while for PEX, it was only 3.03 ± 0.07 mS/cm. On the other hand, there was no significant difference (*p* > 0.05) in the electrical conductivity of the CEX and PEX solutions at 30%. Fonseca et al. [12] reported similar values of electrical conductivity in electrospinning solutions of corn starches with different amylose contents aged for 24 and 48 h. In Hylon V starch, these authors reported values of 2.83 mS/cm and 3.03 mS/cm, and for Hylon VII starch the values were 2.82 mS/cm and 3.01 mS/cm.

### 2.5. Morphological Characterization and Size Distribution of Electrospun Fibers

Table 3 and Figure 5 show the average diameter and morphology of the electrospun fibers at different concentrations of PEX and CEX. In Figure 5g,h, mixtures of fibers and beads (at a similar proportion) are observed, without significant difference (*p* > 0.05). The average diameter of the cassava fibers (CF) was 122 ± 31 nm, while the average diameter for the pea fibers (PF) was 101 ± 26 nm (Table 3). The presence of beads may be due to weak molecular-chain entanglement, generated by low polymer concentration and the insufficient viscosity (Figure 3a,b) of ~0.72 Pa·s and ~1.15 Pa·s in solutions of electrospinning for CEX and PEX. However, when the concentration of CEX and PEX was increased to 34% in the solutions, an increase in fiber-diameter distribution and a viscosity of ~1.42 Pa·s and ~2.01 Pa·s, respectively, was observed; this permitted the production of homogeneous and continuous fibers with random orientation and without beads (Figure 5c,d), with a significant difference (*p* < 0.05) in the average diameter of 344 ± 79 nm and 170 ± 49 nm, respectively. The latter may possibly be attributed to a higher number of chains and a higher molecular weight, responsible for molecular entanglement in CEX.

Likewise, in Table 3, it was observed that the average diameters of PF and CF obtained from the concentrations of 30% and 36% of the PEX and CEX starches presented a significant difference (*p* < 0.05), indicating that the concentration exerted an influence on the average diameter of the fibers and on the reduction of the beads. Furthermore, it was observed that the fiber-diameter distribution broadened with an increasing concentration, regardless of the starch source. Jia et al. [38] observed a similar behavior: as the concentration of poly/(vinyl alcohol) in chitosan solutions increased, nanofibers with the presence of beads changed to a uniform structure without beads.

In another study [13], the authors reported fiber diameters of 300 nm and 150 nm for solutions of Hylon VII starch in formic acid at 100% and 90% *v/v*, values very similar to those obtained in our research. Continuous fibers without beads were obtained in the CEX solutions at 32%, 34%, and 36% *w/v* (Figure 5a,c,e), with an average diameter of 394 ± 102 nm, 344 ± 79 nm, and 316 ± 65 nm, respectively. This result was due to the high content of amylose present in the CEX concentrations and a lower viscosity compared to PEX, allowing for a higher molecular entanglement within a range of viscosities for fiber formation, and increasing the fiber diameter as the concentration increased. The same trend was found in zein solutions: as the zein concentration increased, the fiber diameter increased [39].

In Table 3, it was also observed that concentrations at 32%, 34%, and 36% *w/v* for CEX and PEX did not present a significant difference (*p* > 0.05) in the average diameter of the electrospun fibers. However, in PEX it was observed that defects were generated in the fibers on increasing the concentration to 36%, due to an increase in viscosity that prevented the electrospinning jet from being maintained stable.

### 2.6. Fourier Transform Infrared Spectroscopy (FT-IR)

The FT-IR spectra of NCS, NPS, CEX, and PEX starches and CF and PF electrospun fibers are presented in Figure 6. CEX, PEX, CF, and PF materials exhibited the characteristic spectra of starches (NCS and NPS), suggesting that they exhibited the same type of molecular vibrations without altering the functional groups of the native starch. Assuming that the OH groups of the water molecule absorb energy (3000–3600 cm^−1^), the bands observed in the region of 3294 cm^−1^ correspond to the stretching modes of the OH groups [40,41]. The bands that appear at around 2927 cm^−1^ indicate the existence of methyl groups in the starch chains [42,43]. Starches demonstrate that *C*-*H* stretching modes in the region of 2800–3000 cm^−1^; the different intensities located between 2800 and 3000 cm^−1^ can be attributed to the variation in the amounts of amylose and amylopectin contained in the starch [43]. Therefore, the intensity in the bands suggests a higher amount of amylose in CEX and PEX starches than in NCS and NPS, as a result of extrusion [44].

The vibrational bands of flexion and deformation, related to carbon and hydrogen atoms, can be observed in the region of 1500–1300 cm^−1^. The 1363 cm^−1^ band is attributed to the *C*=*O* and *O*-*H* deformations. The band observed at 1638 cm^−1^ could be assigned to water absorbed in the amorphous regions of the starches. The small differences in the location and intensity of these bands can be attributed to REX causing a loss in molecular weight, especially in amylopectin, as well as a loss of crystallinity and consequently, an increase in the quantity of amylose molecules. The 1712 cm^−1^ band suggests weaker inter- and intramolecular hydrogen bonds, probably due to strong *C*=*O* stretching on the carboxylic groups of the starch formed during electrospinning [45]. The infrared bands observed in the 900–1000 cm^−1^ region, with the appearance of vibrational peaks at 996 and 928 cm^−1^, are related to the vibrations that originate in the *C*-*O*-*C* of the glycosidic bond. The subtle changes in peak location and intensity of the glycosidic-bond band can be attributed to the presence of an α-1,6 bond in amylopectin that shifts the band at higher wavenumbers [40].

The bands at 1149, 1078, and 1017 cm^−1^ are attributed to the contribution of two main vibrational modes *C*-*O* and *C*-*O*-*H*, stretching in the glucose monomer [40,46].

### 2.7. X-ray Diffraction (XRD)

XRD diffraction patterns have permitted the quantification of the amorphous/crystalline relationship in starches and the characterization of the crystalline phases type A, B, and C, the latter being more complex due to the combination of phases A and B [47,48]. Figure 7a shows that NCS presented XRD type A patterns; this indicates a degree of polymerization in ranges from 23–29 of the average length of the amylopectin branched chain. Its branched amylopectin chains are in the form of double helices organized in an orthorhombic structure [49], with a crystallinity value of 18.10% and with predominant peaks at diffraction angles (2θ) of 15.08°, 17.25°, 18.00°, 23.02° and weak reflection at 2θ of 26.66°. These results are in agreement with what has been reported [47], while the NPS (Figure 7b) exhibited type C XRD patterns, indicating a degree of polymerization of 15–17 of the average length of the branched chain [48]. This is characteristic of legumes, with a proportion (crystalline phases A and B) closest to phase A, with a crystallinity value of 18.02%, and with predominant peaks at diffraction angles (2θ) 15.14°, 17.25°, 23.02°, and weak reflections in 2θ of 18.20° and 26.66° [50]. In this respect, the modification in type C starches by REX is more complicated than that in type A starches [48].

On the other hand, we know that native starches possess poor solubility in aqueous solvents, due to the strong hydrogen bonds between starch chains and to their semi-crystalline nature. The REX process promoted the creation of amorphous structures in CEX and PEX (Figure 7a,b), favoring the properties of solubility and viscosity necessary for electrospinning. Furthermore, CF and PF (Figure 7a,b) did not demonstrate recrystallization, which resulted in a network of fibers with an amorphous tendency.

## 3. Materials and Methods

### 3.1. Materials

Formic acid (purity ≥ 95%) was purchased from Sigma-Aldrich Chemical Co. (Toluca, Mexico City, Mexico), NCS (CAS 9005-25-8) was obtained from Ingredion México, S.A. de C.V., and NPS was isolated from dehydrated pea seeds (13% moisture) acquired at a local market (Querétaro, México) according to the methodology reported by Simsek et al. [51]. To obtain the CEX and PEX starches, distilled water was sprayed on the NCS and the NPS starch, respectively, to achieve a moisture level of 23%. The samples were mixed for 5 min to allow for moisture distribution, and the starches were stored in polyethylene bags and allowed to equilibrate for 24 h before their use in an extruder designed and manufactured in the Organic Materials Processing Laboratory of Cinvestav, Querétaro, Mexico. The extruder had a barrel (20-mm internal diameter), a screw (428 mm in length and a 19-mm diameter), three heating zones, and a 4-mm diameter round die. The spindle ratio utilized was 3:1 and the speed was constant at 40 rpm. The temperatures in the feeding, transition, and extrusion zones of the die were maintained constant at 65 °C, 130 °C, and 170 °C, respectively. Subsequently, the CEX and PEX starches were brought to constant weight at 40 °C in a conventional oven (Felisa, model FE-291, Zapopan, Jalisco, Mexico) and ground in a hammer mill (Pulvex, model 200, Mexico City, Mexico) with integrated mesh (150 μm).

### 3.2. Physicochemical Properties of Starches

The AAC, the S, and the SP were determined for the starches NCS, NPS, CEX, and PEX. The AAC was calculated by the iodine colorimetric method [52]. The S and SP were obtained according to Equations (1) and (2) [53].
(1)$$S\;\left(\%\right)=\left(A/W\right)*100$$
(2)$$\mathrm{SP}=\left(P\;*\;100\right)/W\left(100-\;S\right)$$
where *A* refers to the constant weight of the supernatant (g), W represents the weight of the sample (g), and P indicates the weight of the precipitate after centrifugation (g).

### 3.3. Particle Size Distribution

PSD was determined in CEX and PEX by the dynamic light scattering method, using a Zetasizer (Nano-ZS, Malvern Instruments, Ltd., Malvern, UK). The CEX and PEX starches were suspended in distilled water and mixed with a vortex. Particle-refractive and -absorption indices of 1.3 and 0.01, respectively, were employed. The dispersion index of water was 1.33.

### 3.4. Preparation of Starch Solutions

Solutions of the CEX and PEX starches were prepared at concentrations ranging from 0.1–38% *w/v* in aqueous formic acid at 38% *v/v* with stirring at 1000 rpm on a hot plate (Super-Nuova Multi-place, Thermo Scientific, Waltham, MA, USA) at 85 °C for 15 min until homogeneous solutions were obtained.

### 3.5. Rheological Characterization and Electrical Conductivity of Starch Solutions

Starch solutions were measured in a stress-controlled rheometer (ARES, TA Instruments, New Castle, DE, USA) with 16-mm diameter couette geometry and a 10-mm gap. The solutions were poured into the measurement system and allowed to stand for 5 min to equilibrate the temperature at 25 °C. Then an increasing shear rate ranging from 0.1–100 s^−1^ was applied to obtain the flow curves; that is, apparent viscosity (*η*_app_) vs. shear rate.

Moreover, the specific viscosity (*η*_sp_) was calculated with Equation (3), considering the shear viscosity of 0.1 s^−1^ as near the shear rate of the electrospinning process. The resulting values were plotted against the concentration of each starch, and the Ce was determined from the intersection of the fitted slopes in the semi-dilute unentangled and semi-dilute entangled regions. The rheological parameters were analyzed according to the Herschel–Bulkley model (Equation (4)), for the semi-dilute entangled regime, considered as the electrospinning region of CEX and PEX starch solutions due to its significant high degree of entanglement and its influence in determining the Ce in the electrospinning process.
(3)$$\eta_{sp}=(\eta_{0}-\eta_{s})/\eta_{s}$$
where *η*_sp_ is specific viscosity, *η*_0_ is the shear viscosity (Pa·s) of the polymer solutions at 0.1 s^−1^, and *η*_s_ is the viscosity of the solvent (Pa·s).
(4)$$\tau =\tau_{0}+K{\left(\gamma ˙\right)}^{n}$$
where τ is the shear stress (Pa), τ_0_ is the yield stress (Pa), K is the consistency coefficient (Pa·s*^n^*), γ˙ is the shear rate (s^−1^), and *n* is the flow behavior index (-). The electrical conductivity of the electrospinning operating-region solutions was determined according to Ledezma-Oblea et al. [54].

### 3.6. Electrospinning

A horizontal-configuration electrospinning apparatus consisting of a voltage source (Bertan 230 series, Hauppauge, NY, USA), a syringe feed pump (KD Scientific, Holliston, MA, USA), and an aluminum plate as a grounded collector was utilized. CEX and PEX starch solutions that were within the entanglement semi-dilute entangled regime were chosen as electrospinning solutions. Each solution was individually loaded into a 5 mL syringe (Plastipak BD, Mexico City, Mexico) with a 21-gauge blunt needle as a spinneret. The feed rate was set at 0.2 mL/h, the distance between the needle tip and the collector was 15 cm, and the voltage was 19 kV. The electrospinning process was carried out at room temperature (23 ± 2 °C). CF and PF fibers were stored at room temperature until their characterization.

### 3.7. Characterization of Electrospun Fibers and Starches

Fibers (CF and PF) and starches (NCS, NPS CEX, and PEX) were analyzed by SEM, FT-IR, and XRD.

The morphological characterization of the samples was analyzed by SEM (ESEM Philips, XL30, Tokyo, Japan) at an accelerating voltage of 10 kV. Previously, the powders of the starches and fibers were placed and fixed directly on a metallic support with double-sided carbon tape and covered with gold/palladium in a vacuum chamber (Denton Vacuum in Sputtering DESK V, Moorestown, NJ, USA). Fiber-diameter distribution and starch-granule sizes were determined for each micrograph using ImageJ 1.53k (National Institutes of Health, Bethesda, MD, USA).

The samples were analyzed by FT-IR with the attenuated total reflection (ATR) technique employing a spectrometer (Perkin Elmer, Spectrum GX, Inc., Waltham, MA USA). Each sample was held in a diamond crystal of the instrument and 24 scans were recorded for each spectrum, with a wave-number resolution of 4 cm^−1^. Spectra were measured in the wave-number range of 600–4000 cm^−1^.

For the XRD analysis, an X-ray diffractometer (Rigaku, DMAX-2100, Tokyo, Japan) was used, each sample was placed in a glass sample holder, and the analysis was performed with a Bragg scanning angle of 0–45° by scanning 2θ; with CuKα radiation, the irradiation conditions were at a voltage of 40 kV with scanning-speed intervals of 0.05 2θ/min.

### 3.8. Statistic Analysis

Measurements of ACC, SI, SP, rheological parameters, and electrical conductivity were performed in triplicate, and the average diameters of the electrospun fibers were evaluated using 50 randomly selected fibers. All data were statistically analyzed by analysis of variance (ANOVA) with Tukey test. Significant differences were determined with a 95% confidence interval (95% CI) (*p* < 0.05), expressed as mean ± standard deviation (SD).

## 4. Conclusions

In this study, the application of REX in cassava and pea starches improved the properties of solubility, viscosity, and amylose content in electrospinning solutions; as a result, there was a successful manufacture of continuous fibers without beads with an average diameter of 316 ± 65 at 394 ± 102 nm for CF and at 170 ± 49 nm for PF with amorphous structure and characteristic functional groups of starches. We observed that the content of amylose in the electrospinning solutions comprises an important parameter that defines the diameter of the fibers and the viscosity of the solutions. Rheological analysis of the electrospinning solutions indicated that to obtain well-formed fibers, the concentration of CEX and PEX must be 1.14 and 1.25 times the Ce, respectively. Therefore, the experimental results revealed that REX as a pretreatment can potentially be used to improve the attributes of cassava and pea starch for obtaining fibers by electrospinning, thus expanding its possible use in food and non-food applications as vehicles of administration of bioactive compounds.

## Figures and Tables

**Figure 1 molecules-27-05944-f001:**
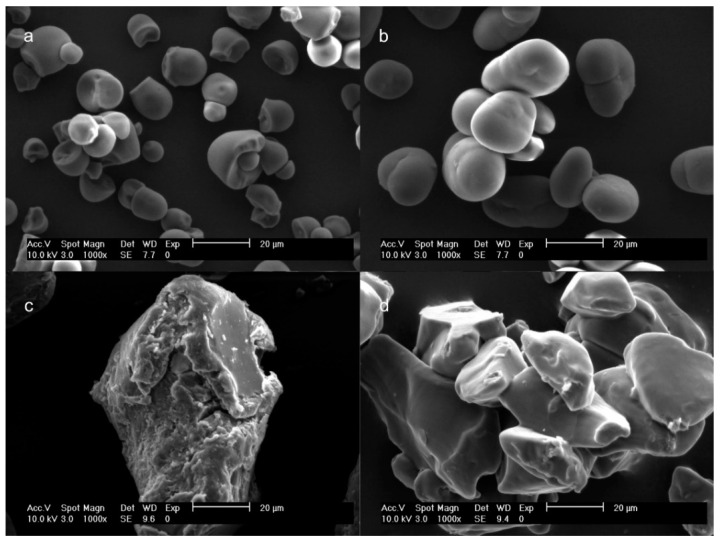
SEM micrographs of native and extruded starches: (**a**) native cassava starch (NCS); (**b**) native pea starch (NPS); (**c**) extruded cassava starch (CEX); and (**d**) extruded pea starch (PEX).

**Figure 2 molecules-27-05944-f002:**
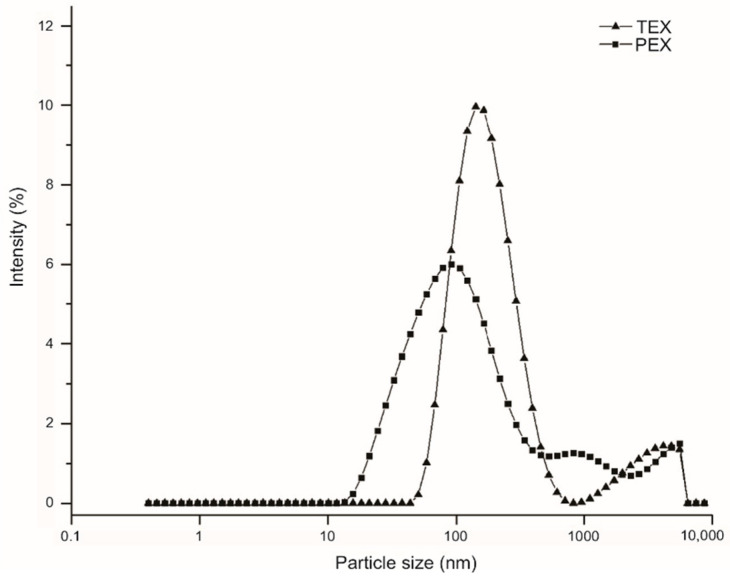
Particle-size distribution (by intensity %) of the extruded starches: extruded cassava starch, (CEX) and extruded pea starch (PEX).

**Figure 3 molecules-27-05944-f003:**
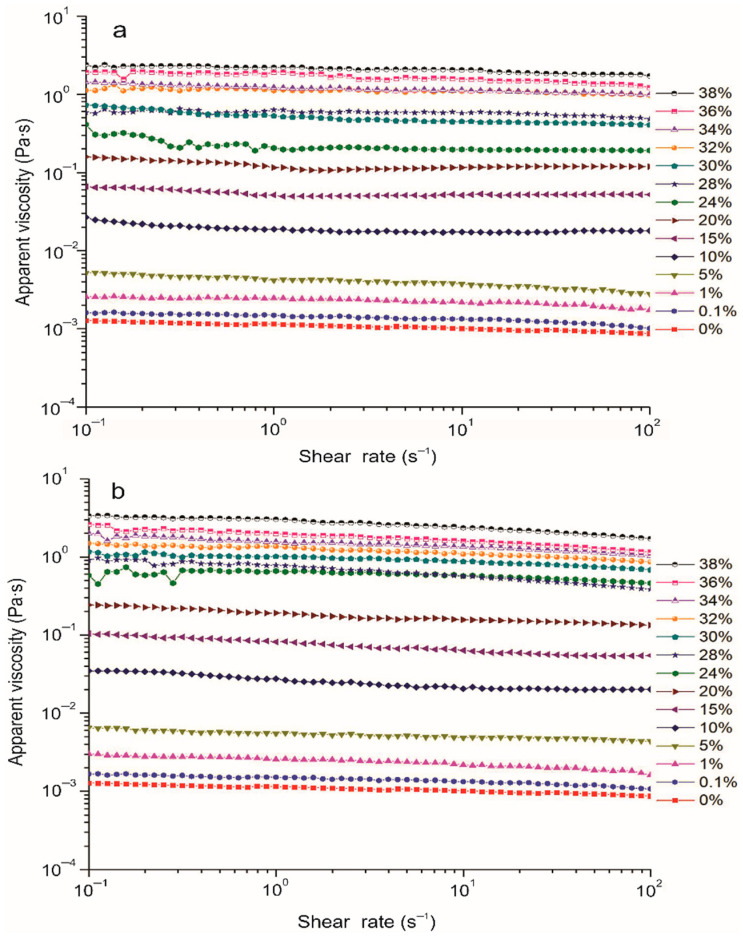
Flow curves of extruded starch in aqueous formic acid (38% *v*/*v*) as a function of concentration (*w*/*v*) at 25 °C: (**a**) extruded cassava starch (CEX) and (**b**) extruded pea starch (PEX).

**Figure 4 molecules-27-05944-f004:**
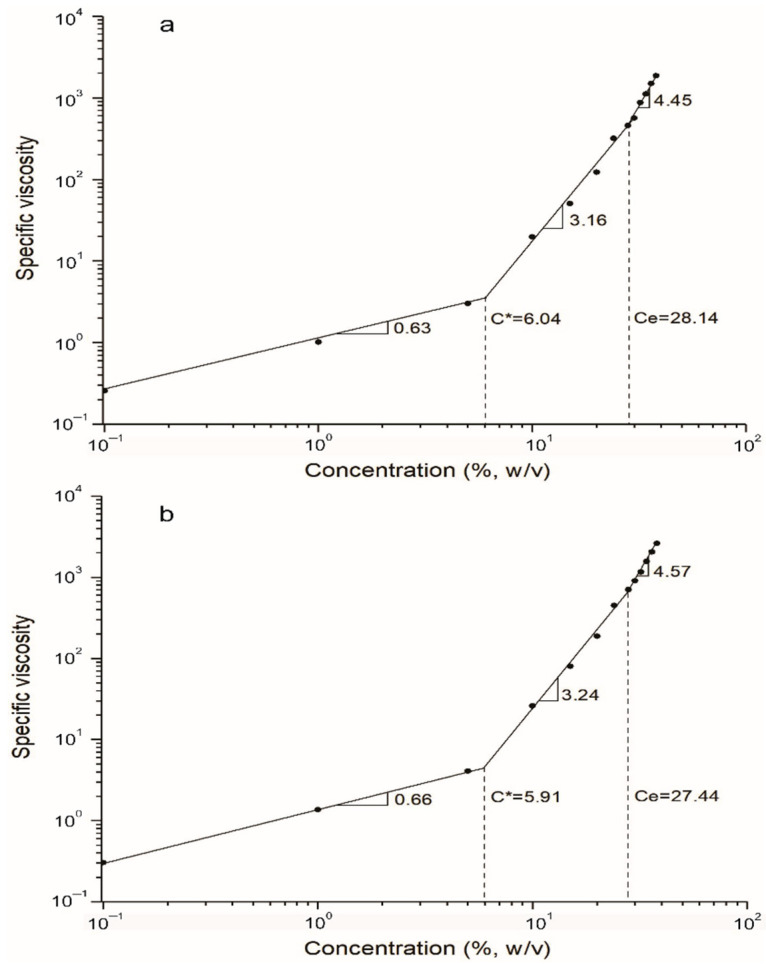
Plot of specific viscosity vs. starch concentration (0.1–38% *w*/*v*) in aqueous formic acid 38% (*v*/*v*): (**a**) extruded cassava starch (CEX) and (**b**) extruded pea starch (PEX). The overlap concentration (C*), the entanglement concentration (Ce), and the slopes of the fitted lines in three regimes are illustrated.

**Figure 5 molecules-27-05944-f005:**
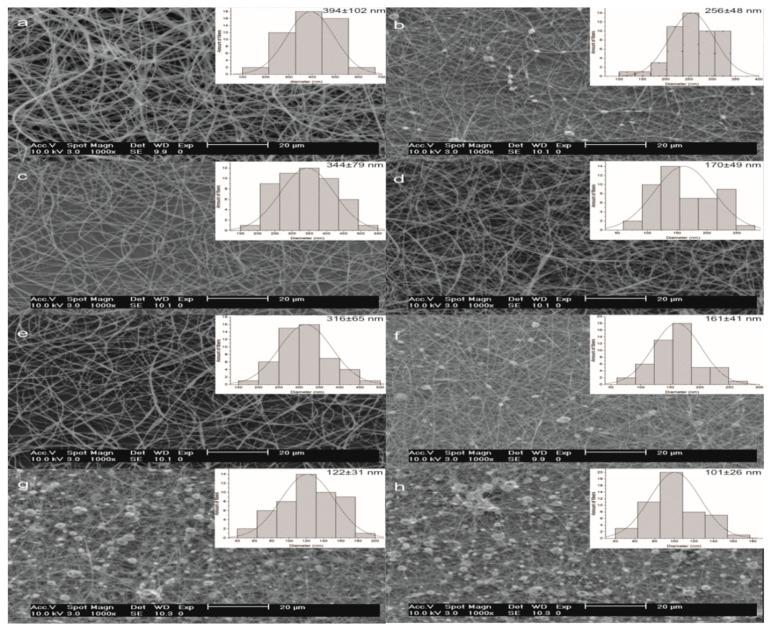
SEM micrographs and histograms with the average diameter of electrospun fibers from extruded starches at different concentrations. Extruded cassava starch (CEX): (**a**) 36%; (**c**) 34% (**e**) 32%, and (**g**) 30%, and PEX extruded pea starch: (**b**) 36%; (**d**) 34%; (**f**) 32%, and (**h**) 30%.

**Figure 6 molecules-27-05944-f006:**
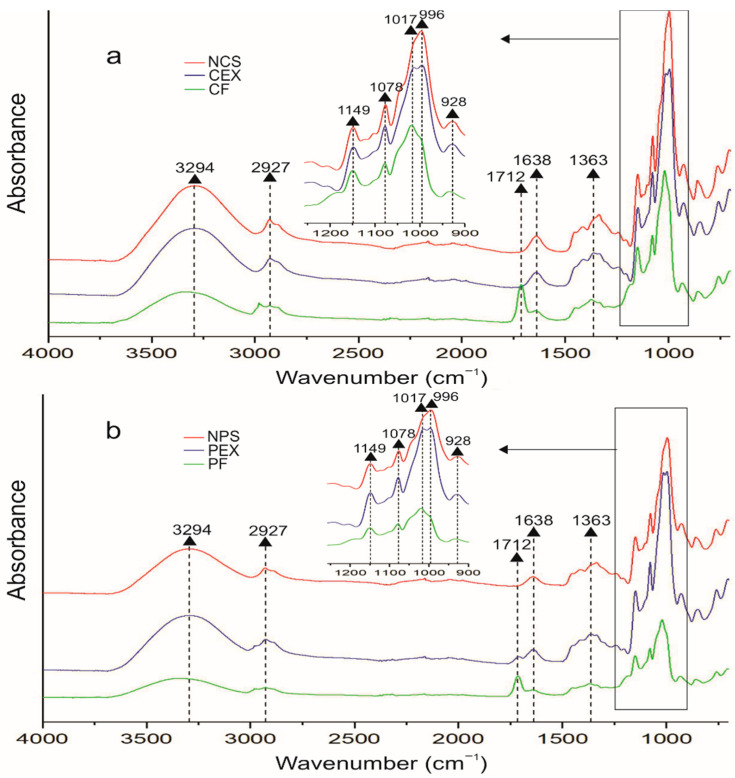
FT-IR spectra of starches and electrospun starch fibers: (**a**) native cassava starch (NCS), extruded cassava starch (CEX), and cassava fibers (CF) and (**b**) native pea starch (NPS) extruded pea starch (PEX) and pea fibers (PF).

**Figure 7 molecules-27-05944-f007:**
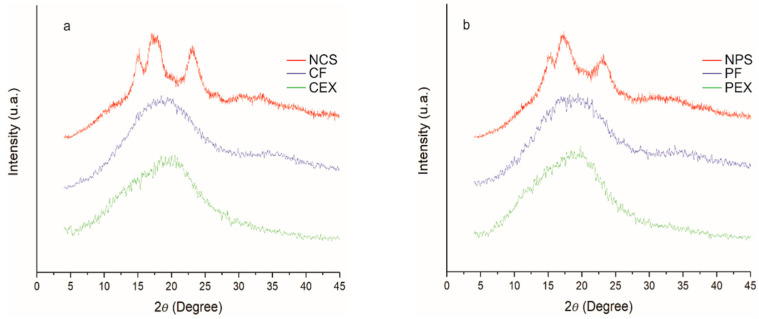
XRD patterns of starches and electrospun starch fibers: (**a**) native cassava starch (NCS), extruded cassava starch (CEX), and cassava fibers (CF), and (**b**) native pea starch (NPS), extruded pea starch (PEX), and pea fibers (PF).

**Table 1 molecules-27-05944-t001:** Apparent amylose content (AAC) and properties of native and extruded starches ^1^.

Starches ^1^	AAC (%)	S (%)	SP (g gel/g Starch)
NPS	32.24 ± 0.12 d	15.02 ± 0.10 d	16.90 ± 0.13 d
NCS	21.41 ± 0.06 c	22.56 ± 0.21 b	22.95 ± 0.16 b
PEX	43.36 ± 0.13 b	21.30 ± 0.13 c	21.77 ± 0.07 c
CEX	49.02 ± 0.13 a	48.12 ± 0.04 a	37.41 ± 0.29 a

^1^ S, solubility; SP, swelling power; NPS, native pea starch; NCS, native cassava starch; PEX, pea extruded starch, and CEX, cassava extruded starch. Assays were performed in triplicate, and results are expressed as mean ± standard deviation (SD). Different lowercase letters in the same column are significantly different (*p* < 0.05) using Tukey test.

**Table 2 molecules-27-05944-t002:** Rheological parameters of solutions at different concentrations of extruded starches.

Rheological Parameters	Concentration (% *w*/*v*)				
	30	32	34	36	R^2^
Yield stress τ_0_ [Pa]					
CEX	0 ± 0.10 aA	0 ± 0.07 aA	0 ± 0.08 aA	0 ± 0.2 aA	0.99
PEX	0 ± 0.04 aA	0 ± 0.05 aA	0 ± 0.06 aA	0 ± 0.08 aA	0.99
Consistency coefficient (K) [Pa∙s*^n^*]					
CEX	0.50 ± 0.00 cB	1.29 ± 0.02 bB	1.33 ± 0.02 bB	2.18 ± 0.08 aA	0.99
PEX	1.14 ± 0.01 dA	1.46 ± 0.02 cA	1.75 ± 0.02 bA	2.37 ± 0.03 aB	0.99
Flow behavior index (*n*) [-]					
CEX	0.95 ± 0.00 aA	0.94 ± 0.00 bA	0.94 ± 0.00 bA	0.88 ± 0.01 cA	0.99
PEX	0.89 ± 0.00 aB	0.89 ± 0.00 aB	0.88 ± 0.00 bB	0.85 ± 0.00 cB	0.99

CEX, extruded cassava starch; PEX, extruded pea starch; CF, cassava fibers, and PF, pea fibers. Assays were performed in triplicate, and results are expressed as mean ± standard deviation (SD). Different capital letters in the columns and different lowercase letters in the rows indicate significant differences (*p* < 0.05) using Tukey test.

**Table 3 molecules-27-05944-t003:** Electrical conductivity of solutions of extruded starches at different concentrations and diameter distribution of electrospun starch fibers.

Concentration (% *w*/*v*)	Values of Electrical Conductivity (mS/cm)		Diameter Distribution of Electrospun Fibers (nm)	
	CEX	PEX	CF	PF
30	3.41 ± 0.09 aA	3.46 ± 0.33 aA	122 ± 31 aB	101 ± 26 aB
32	2.97 ± 0.27 aAB	3.23 ± 0.32 aAB	316 ± 65 aA	161 ± 41 bAB
34	2.97 ± 0.09 aAB	3.03 ± 0.07 aAB	344 ± 79 Aa	170 ± 49 bAB
36	2.77 ± 0.20 aB	2.58 ± 0.24 aB	394 ± 102 aA	256 ± 48 aA

CEX, extruded cassava starch; PEX, extruded pea starch; CF, cassava fibers, and PF, pea fibers. Assays were performed in triplicate, and results are expressed as mean ± standard deviation (SD). Different capital letters in the columns and different lowercase letters in the rows indicate significant differences (*p* < 0.05) using Tukey’s test.

## Data Availability

Not applicable.

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
