# Peer review of "Reactive Extrusion as a Pretreatment in Cassava (Manihot esculenta Crantz) and Pea (Pisum sativum L.) Starches to Improve Spinnability Properties for Obtaining Fibers"

_molecules, 2022, doi:10.3390/molecules27185944_

Round 1
Reviewer 1 Report
Comments
This manuscript “Reactive extrusion as a pretreatment in cassava (Manihot escu- lenta Crantz) and pea (Pisum sativum L.) starches to improve spinnability properties for obtaining fibers” is interesting and has potential utility in deciding the role of starch in the preparation of various food products. Some questions in the 'comments to the authors' section need clarification. These are just a bit more than 'minor revisions', so I must select 'major revisions' for this first recommendation.
1. Abstract: What is the purpose? What is the conclusion?
2. Add more information about cassava and pea starches.
3. At the end of the Introduction section, add objectives and justify why this study was conducted.
4. Add the unit of SP in Table.1. Provide basic reason for the change in solubility and swelling index. Authors are advised to reanalyse the swelling and solubility power, these values look slightly high.
5. Figure 3, re-plot this figure as graph starting lines are not same. This should start from zero, while in these graph shear rate values showed in minus (-).
6. Conclusion is not informative; rewrite this section.
Author Response
We thank you very much for your time and suggestions in revising our manuscript.

Reviewer 2 Report
David Tochihuitl-Vázquez and coworkers report obtaining of electrospun fibers from cassava (CEX) and pea (PEX) starches pretreated by reactive extrusion (REX). And they provide detail discussion of the effect of different parameters on the final electrospun fibers. This work offers the opportunity to expand the use of starches in obtaining electrospun fibers for food and biomedical applications as vehicles for the encapsulation of bioactives. The whole content is carefully as well as logically organized, which would be highly desired for readers in the field of Molecules. Thus, this work could be published in Molecules after minor revision.
1) How to prepare the sample for observing the derived morphology via SEM technology? Is the starch directly placed upon the conductive tape?
2) For increasing the readability of this manuscript, the authors should introduce some presentation about "the application of starch in other different field, such as hydrogel, optoelectronic devices, et al", and then some derived literatures should be cited in the introduction, i.e., Materialstoday Proceedings, 2017, 4, 12238; International Journal of Biological Macromolecules, 2021, 190, 189; Nano Energy, 2020, 76, 104964; Journal of Materials Chemistry A, 2018, 6, 4023.
3) The language should be slightly improved.
Author Response
Le agradecemos mucho su tiempo y sus sugerencias para revisar nuestro manuscrito.

Round 2
Reviewer 1 Report
Authors address all comments suggested by reviewer. Now this manusript is acceptable in this esteem journal.